# Mussel Shell-Derived Macroporous 3D Scaffold: Characterization and Optimization Study of a Bioceramic from the Circular Economy

**DOI:** 10.3390/md18060309

**Published:** 2020-06-12

**Authors:** Stefania Scialla, Francesca Carella, Massimiliano Dapporto, Simone Sprio, Andreana Piancastelli, Barbara Palazzo, Alessio Adamiano, Lorenzo Degli Esposti, Michele Iafisco, Clara Piccirillo

**Affiliations:** 1Institute of Nanotechnology (NANOTEC), National Research Council (CNR), Campus Ecoteckne, Via Monteroni, 73100 Lecce, Italy; stefania.scialla@inl.int; 2Institute of Science and Technology for Ceramics (ISTEC), National Research Council (CNR), Via Granarolo 64, 48018 Faenza, Italy; francesca.carella@istec.cnr.it (F.C.); massimiliano.dapporto@istec.cnr.it (M.D.); simone.sprio@istec.cnr.it (S.S.); andreana.piancastelli@istec.cnr.it (A.P.); alessio.adamiano@istec.cnr.it (A.A.); lorenzo.degliesposti@istec.cnr.it (L.D.E.); 3Ghimas SpA, C/O Ditech S.c.a.r.l., Campus Ecotekne, 73100 Lecce, Italy; barbara.palazzo@enea.it

**Keywords:** bioceramics, scaffolds, hydroxyapatite, circular economy, by-products, mussel shells

## Abstract

Fish industry by-products constitute an interesting platform for the extraction and recovery of valuable compounds in a circular economy approach. Among them, mussel shells could provide a calcium-rich source for the synthesis of hydroxyapatite (HA) bioceramics. In this work, HA nanoparticles have been successfully synthesized starting from mussel shells (*Mytilus edulis*) with a two steps process based on thermal treatment to convert CaCO_3_ in CaO and subsequent wet precipitation with a phosphorus source. Several parameters were studied, such as the temperature and gaseous atmosphere of the thermal treatment as well as the use of two different phosphorus-containing reagents in the wet precipitation. Data have revealed that the characteristics of the powders can be tailored, changing the conditions of the process. In particular, the use of (NH_4_)_2_HPO_4_ as the phosphorus source led to HA nanoparticles with a high crystallinity degree, while smaller nanoparticles with a higher surface area were obtained when H_3_PO_4_ was employed. Further, a selected HA sample was synthesized at the pilot scale; then, it was employed to fabricate porous 3D scaffolds using the direct foaming method. A highly porous scaffold with open and interconnected porosity associated with good mechanical properties (i.e., porosity in the range 87–89%, pore size in the range 50–300 μm, and a compressive strength σ = 0.51 ± 0.14 MPa) suitable for bone replacement was achieved. These results suggest that mussel shell by-products are effectively usable for the development of compounds of high added value in the biomedical field.

## 1. Introduction

The circular economy has been recently proposed as a new paradigm for sustainable development, showing the potential to generate new business opportunities in worldwide economies and to increase resource efficiency in manufacturing significantly. The vision of the circular economy paradigm is to fundamentally change the current linear “take–make–dispose” economic approach, which is the cause of massive waste flows, with the new approach of “reduce, re-use, and retain”. Therefore, nowadays, the use of industrial by-products as starting materials has attracted increasing interest in the extraction and/or procurement of chemical compounds with valuable properties and possible technological applications [1,2]. In particular, the valorization of residues from marine origin, such as those normally obtained from the food industry, could constitute an interesting platform in the circular economy for the development of high added value compounds with financial and environmental advantages. 

Ceramics and even bioceramics are not alien to this paradigm shift. Bioceramics are an important class of ceramics especially designed for the repair and regeneration of some damaged portions of the human body. Bioceramics exhibit chemical and mechanical features mimicking those of the human calcified tissues; because of this, many studies have been devoted to the use of these materials for bone and dental replacement and regeneration [3]. Possible promising applications of bioceramics for soft tissue regeneration have also been emerging in recent years [4]. 

Examples of bioceramics currently employed in clinical applications include dense bioinert components (e.g., middle ear ossicles or load-bearing components of joint prostheses), sintered porous scaffolds, granules for bone filling, coatings on metal joint prostheses, as well as injectable formulations (i.e., cements) [5,6]. Among them, porous scaffolds are one of the most promising devices in tissue engineering, and they are designed as three-dimensional templates able to support cell adhesion and proliferation, while also favoring the physiological regeneration of bone tissue [7,8]. The mechanical and biological performances of porous scaffolds are greatly affected by chemical composition and porosity. The ideal macroporous size distribution to achieve optimal cellular growth is still subject to intense debate, but several evidences have shown that pores with a diameter between 100 and 500 μm favor cell proliferation [9]. Moreover, the presence of small and interconnected pores (0.1–1.0 μm) seems also to play a crucial role [10]. In this context, the selection of appropriate preparation methods to obtain scaffolds with specific pore size and structure appears crucial [11].

Several methods have been used to manufacture 3D bioceramic scaffolds [12,13,14]; among these, the direct foaming technique is considered particularly promising as it allows the preparation of a porous 3D structure without using any sacrificial template [15]. Pore formation is achieved through the incorporation of a gas (usually air) in a ceramic suspension/slurry; subsequently, the suspension is poured, dried, and sintered [16]. The extent of porosity and the pore dimension are primarily determined by the amount of gas incorporated into the slurry and the stability of the slurry itself, respectively. These characteristics can be tailored using appropriate surfactant(s) and/or foaming agent(s). 

Calcium phosphates in the form of hydroxyapatite (HA) (Ca_10_(PO_4_)_6_(OH)_2_) are ideal materials for the preparation of 3D porous scaffolds for bone tissue engineering. Being a chemical close to the mineral phase of bone, HA is intrinsically endowed with the excellent properties of bioactivity, biocompatibility, and osteoconductivity [17]. Literature reports the use of different food industry by-products, such as eggshells [18] and seashells [19,20], as starting material (calcium source) for the production of HA or other calcium phosphates. The main component of shells is calcium carbonate (CaCO_3_), which can be converted into CaO by thermal treatment and then into HA by a reaction with an appropriate phosphorus source. Mussel shells are one of the most abundant by-products produced by fish industries. Despite their potential as valuable compounds source, they are not sufficiently valorized—indeed, in most cases, they are simply disposed of. As with other seashells, mussel shells can also be employed to prepare HA-based materials; however, very few studies investigating this preparation are reported in the literature [21,22,23,24]. Moreover, these works are limited to the synthesis of HA in powder form, and none of them report on the preparation of 3D structures made from mussel-derived powders. 

In this work, we report the synthesis of HA based on thermal treatment of mussel shells to convert CaCO_3_ to CaO followed by a wet chemical precipitation with two different phosphorus sources (Figure 1). Tests were performed to select the most appropriate experimental conditions for the thermal treatment of the mussel shells as well as to generate HA further. 

Once the conditions were optimized, the process of synthesis of HA was scaled up at the pilot level, and the obtained powder was employed to fabricate porous scaffolds using the already reported direct foaming method (Figure 2) [25]. This method has also been selected due to its low economic and environmental impact since it does not use sacrificial templates for obtaining scaffolds. Both powders and 3D structures were fully characterized, in particular, analysis of the porosity and the mechanical properties of the scaffolds was also performed.

## 2. Results and Discussion

### 2.1. Preparation and Characterization of CaO Samples from Mussel Shells

The X-ray diffraction (XRD) pattern of raw mussel shells (Appendix A) and the relative phase quantification by Rietveld refinement confirmed the presence of only the CaCO_3_ phase in the form of calcite (70 ± 1 wt%) and its polymorph aragonite (30 ± 1 wt%). Thermogravimetric analysis (TGA) curve of raw mussel shells (Appendix A) displays two main weight losses, the first of about 4% in the range 250–600 °C attributed to the removal of organic matter, the second of about 42% in the range 670–900 °C attributed to the decarbonization of CaCO_3_ [26,27]. A multi-elemental analysis of the shells by inductively coupled plasma atomic emission spectrometer (ICP-OES) (Appendix A) revealed that the most abundant elements among those detected were Ca, Mg, P, Na, Se, and Sr. The concentrations of the other 15 elements (Appendix A) were below their detection limit.

Table 1 reports the preparation conditions of the CaO based samples derived from mussel shells. Two different temperatures (700 or 1000 °C) were considered for the thermal treatment; the lowest value was selected as the minimum temperature essential to ensure the complete removal of the organic fractions from the shell which is generally up to 5 wt% [26], while the highest was considered to achieve the complete decomposition of carbonates [27]. Moreover, to compare the effect of the gaseous atmosphere on the composition and the specific surface area (SSA) of the samples, thermal treatments were performed in N_2_ or air.

XRD patterns of the obtained powders as well as of the standard CaO, CaCO_3_ and Ca(OH)_2_ are shown in Figure 3, while the corresponding phase composition evaluated through Rietveld refinement is reported in Table 1. 

The XRD pattern of S_A_700 (air, 700 °C) showed sharp and intense peaks at 2θ = 37.35 and 53.85° ((200) and (220), respectively) belonging to CaO as a result of the decarboxylation of CaCO_3_ occurring during the thermal treatment. A small amount of CaCO_3_ was still detectable in the diffraction pattern (2θ = 29.41°, (104)), indicating that the carbonate decomposition was not complete. Larger and less defined peaks at 2θ = 18.06 and 34.14° ((001) and (101), respectively) belonging to calcium hydroxide Ca(OH)_2_ were also present, ascribable to the reaction of CaO with the atmospheric moisture. Indeed, as shown in Table 1, Ca(OH)_2_ represents the main phase for S_A_700. Increasing the temperature to 1000 °C (sample S_A_1000), CaO represented the only phase present. 

With the thermal treatment under N_2_ at 700 °C (S_N_700), the peaks attributed to CaO and CaCO_3_ showed similar intensities; the phase analysis (Table 1) indeed showed that the two compounds were present in comparable relative proportions; this means that, in an inert atmosphere, the decarboxylation takes place at a lower extent in comparison to the treatment in air. Ca(OH)_2_ traces were also detected, but at a lower proportion than the sample treated in air. Using N_2_ at 1000 °C (sample S_N_1000), a decrease in the CaCO_3_ peaks took place, indicating a more enhanced, although incomplete, decarboxylation. These differences in the composition of the samples prepared in diverse atmospheres can be explained considering the different mechanisms and kinetics of shells’ decomposition under either air or N_2_. For what concerns the shell organic component behavior upon treatments, it is known that in an inert atmosphere, the organic carbon is converted into inorganic graphitic-like carbon [28]. Moreover, literature reports that the atmosphere gas can affect the capability of the carbon dioxide to diffuse through the porous structure of the mussel shells [29]; such diffusion, in fact, is higher in air than in N_2_ and could lead to an increase in decarboxylation extent. Similar to the raw mussel shells in all the samples, the most abundant elements among those detected were Ca, Mg, P, Na, Se, and Sr (Appendix A). In agreement with the phase composition, S_A_1000 had a slightly higher content of Ca with respect to the other samples. 

Scanning electron microscopy (SEM) micrographs of the powders treated in different conditions are shown in Appendix A. S_A_700 (Appendix A) shows rounded shaped structures, aggregates with dimensions up to micron scale. At higher temperatures (S_A_1000, Appendix A), structures become larger. For samples treated in N_2_ (Appendix A), different morphology could be observed; the surface of both samples, however, showed smaller lamellae-like features with some cracks between them (visible at higher magnification). 

Table 1 reports the values of the SSA for the above-mentioned samples; it is possible to observe that the SSA slightly decreased by increasing temperature—from 5.6 to 3.1 m^2^/g—for the samples treated in air. On the other hand, an opposite trend was observed—from 2.5 to 6.6 m^2^/g—for the ones treated with N_2_. Comparing the samples prepared in air with those prepared in N_2_, the inert atmosphere showed to be less suitable for the conversion process of CaCO_3_ in CaO, as led to powders with a relatively high content of calcite; because of this, the treatment in N_2_ was not considered further in this study. It is also worth noting that environmental and economic sustainability always has to be considered when working with by-product valorization in a circular economy approach [30]. In this context, the use of air was also preferred with the aim of developing the simplest process, with the lowest cost and impact on the environment. 

### 2.2. Preparation and Characterization of HA Samples

Samples S_A_700 and S_A_1000 were used to prepare HA using two different protocols, as detailed in Section 3.3 and summarized in Table 2; the XRD data for these samples as well as of standard HA are reported in Figure 4a,b. 

It can be seen that HA as a single crystalline phase was obtained in all synthesis, apart from HA_2, which showed traces of calcite (residual unreacted starting material). No other calcium phosphate phases were present. ICP-OES analysis of the samples showed that in all powders, the Ca/P ratio was slightly higher than the stoichiometric one (i.e., 1.67), around 1.73 for the powders prepared with H_3_PO_4_ and 1.88 for those with (NH_4_)_2_HPO_4_ (Table 2).

Differences in the crystallinity of the powders can be observed; in fact, samples prepared using (NH_4_)_2_HPO_4_ as phosphorus source were more crystalline than those obtained using H_3_PO_4_. More specifically, XRD pattern of HA_1 showed the sharpest peaks; moreover, the signals at 31.7 and 32.2° were clearly distinct and separate (Figure 4b). For HA_3, although overall peaks were broader, those at 31.7 and 32.2° were still separated. For HA_2 and HA_4, on the other hand, these latter two peaks were overlapped into a single broad one indicating a very low crystallinity degree. The difference in crystallinity between the samples prepared with the H_3_PO_4_ and those with the (NH_4_)_2_HPO_4_ can be explained considering the different mechanisms according to which the reaction for HA formation takes place (see Equations (3) and (4)). In fact, the overall and local pH (about 9) at which the HA nucleation occurs is higher when (NH_4_)_2_HPO_4_ is used as a phosphorus source with respect to what happens using H_3_PO_4._ In the second case, the reaction initial pH (about 9), decreases down to less than 7 during the H_3_PO_4_ dropping, because the reaction goes through a neutralization mechanism. Nucleation occurring at neutral or slightly acid condition gives rise to a less crystalline HA than that obtained at basic pH, other conditions being equal.

The average size of crystal domains along the *c*-axis (D_(002)_) and the *a–b* plane (orthogonal to the *c*-axis, D_(310)_), calculated by Scherrer’s equation, using the width at half height for the respective reflections (002) and non-overlapped (310), are shown in Table 3. D_(002)_ and D_(310)_ of HA_1 and HA_3 that were prepared with (NH_4_)_2_HPO_4_ were both larger than the corresponding crystal domains of HA_2 and HA_4 (prepared with H_3_PO_4_). These data also revealed that all the crystals were elongated along the *c*-axis with HA_3 having the maximum aspect ratio, estimated as D_(002)_/D_(310)._

The bigger crystal domains’ size of the sample HA_1 corresponded to a lower value of SSA, as shown in Table 2; smaller crystal domains’ size, on the other hand, led to higher SSA, especially for samples HA_2 and HA_4 (about 90–100 m^2^/g). 

Differences in the HA samples could also be seen by TEM micrographs; HA_3 (Figure 4e), in fact, constituted needle-like nanoparticles with a more elongated shape, up to about 100 nm. TEM images of HA_1, HA_2, and HA_4 (Figure 4c,d,f, respectively), on the other hand, showed nanoparticles with a plate shape morphology. SEM images (Appendix A) of HA samples showed morphologies similar to those observed with TEM microscopy. 

These results showed that selecting the appropriate mussel shells’-derived materials and/or the reaction conditions, it is possible to tailor the characteristics of HA samples; the use of the obtained HA powders will be different according to their properties/features. For applications in environmental remediation or drug delivery, for instance, powders with high SSA (such as HA_4) would be more suitable [31,32]. To make 3D scaffolds, on the other hand, the interactions between the particles must be considered. Therefore, for some protocols, the use of powders with larger SSA may not lead to good results [33]. For the specific application of the present study, as the foaming method was employed, sample HA_1 was chosen as the most suitable, having the lowest SSA. Therefore, the synthesis conditions to obtain HA_1 were scaled up by 100 times, as described in Section 3.3 (sample HA_5). The XRD pattern (Appendix A) of HA_5 and the relative phase quantification by Rietveld refinement revealed that the powder was constituted by HA as the major phase with a very low content of calcite residue (calcite < 0.5 wt%). Comparing the XRD patterns of HA_1 and HA_5, it can be seen that peaks were slightly less sharp for the sample prepared on a larger scale; this finding is in agreement with the crystal domains’ size of HA_5, that were slightly lower than HA_1 (Table 3). The SSA of HA_5 was 63.8 m^2^/g, which is an intermediate value coherent with the trend observed for the crystal domains’ size. The SEM micrograph of sample HA_5 is reported in Appendix A. It shows the presence of rounded granules, smaller than 100 nm; a similar morphology was observed for sample HA_1 (Appendix A), hence confirming that the HA main characteristics are retained even upon the scaling-up process.

In conclusion, the HA_1 preparation process was scaled up successfully, and the HA_5 sample is suitable in terms of SSA to be used for scaffold fabrication by foaming. Indeed, HA_5 had a value of SSA comparable to HA-based materials that were previously used for preparing 3D scaffolds by direct foaming technique [25].

### 2.3. Preparation and Characterization of the 3D Scaffolds

Macroporous 3D scaffolds were prepared using the HA_5 sample, and, as a comparison, a commercial HA powder already used for this preparation [25]. The methodology employed here for the scaffold generation was based on a patented process [34] with a crucial modification; in fact, the use of high-energy planetary ball milling allowed to shorten the process time [25] significantly. Calcination was performed on both powders to further lower SSA and to prevent the formation of agglomerates in the ceramic suspension (slurry) [25]. Characterizations of HA_5 after calcination at 1000 °C (hereafter labeled as HA_6) and commercial HA are reported in Table 4.

Indeed, HA_6 showed a much lower surface area (4.5 m^2^/g) than HA_5, due to the coalescence of the primary particles after thermal activation; moreover, the SEM morphological analysis of this powder showed granules size in the range from 300 to 700 nm (Appendix A). Analysis of the XRD pattern of HA_6 (Appendix A) showed that CaO was also present in the powder (about 4.0 wt%). The HA thermal decomposition to form CaO after high-temperature treatments was previously reported in the literature [35,36]. The formation of CaO is due to the removal of the excess of calcium ions from the crystal lattice; in fact, HA_5 is non-stoichiometric and has a surplus of Ca^2+^ ions, as evinced by the Ca/P molar ratio (i.e., 1.83 ± 0.01) which is higher than the stoichiometric one (i.e., 1.67). In HA_6, a considerable increase in crystallinity was observed by the presence of intense, highly resolved diffraction peaks, and crystal domains’ size was estimated in the order of hundreds of nanometers for both D_(002)_ and D_(310)_ due to grain growth with calcination. As reported in Section 3.4, the scaffold was prepared with a slurry made with Dolapix as a dispersant agent. To assess the stability of the powders in contact with this compound, the distribution of the particle size was measured before and after the calcination—see Appendix A showing dynamic light scattering (DLS) data of dimension vs. intensity for the two suspensions. The HA_5 sample showed a particle size distribution in the range from 200 to 500 nm, indicating that particles tend to agglomerate into larger aggregates, as previously observed [37]. The calcination of the powder up to 1000 °C (sample HA_6) led to a broader particle distribution ranging from 500 and 1500 nm, also confirmed by SEM images (Appendix A).

The ξ-potential values were negative for both samples, −40 ± 5 mV and −30 ± 03 mV for HA_5 and HA_6, respectively. These values highlighted a significant stability for both HA suspensions analyzed [38,39]. HA_6 powder was used to prepare the scaffold via direct foaming and successively sintered (see Section 3.4). In particular, the foamed suspensions were poured into laboratory filter paper molds and kept motionless at 25 °C for 24 h to ensure a slow drying process to prevent the generation of cracks or flaws in the scaffold. The crystal phase composition was investigated by XRD and showed that the scaffold was composed of HA as the main crystalline component with about 4 wt% of CaO, thus keeping the same composition of the precursor powder (data not shown). SEM micrographs of the scaffold showed (Figure 5a,b) a highly porous structure (estimated in the range 87–89%) with an open and interconnected micro- and macro-porosity in the range of 50–300 μm. This feature makes the scaffold suitable to successfully support the infiltration, migration, and proliferation of the cells, as well as a good perfusion of physiological fluids [7]. The scaffolds prepared with the commercial HA sample used as control (Figure 5c,d), on the other hand, highlighted a higher degree of consolidation. 

The values of the compressive strength of the scaffolds are reported in Table 5. The mechanical performances of porous structures are strongly related to the porosity; in this study, considering the relatively high porosity (about 90% porosity), the compressive strength values were in agreement with the previously reported strength-porosity curves [25]. Comparing with literature data of porous structures of marine origin, Hadagalli et al. [40] reported scaffolds made from cuttlefish bones that were prepared by the foaming method. In this case, the maximum achieved porosity was about 50%, i.e., much lower than that obtained in the present study.

Despite a significant decrease in compressive strength in comparison to the control, the HA_6 scaffolds also exhibited a significantly higher strain toughness (work of fracture, Figure 5e), intended as the energy of mechanical deformation per unit volume before fracture. This can be explained considering the different morphologies shown by the SEM micrographs, which were more porous for the HA_6 scaffold and more consolidated for the control one.

It should be noted that within a porous material, the stress propagation is mainly dominated by the small solid areas, which form walls and edges of the pores. In this condition, the stress is only concentrated in such small regions, and it will lead to plastic or brittle deformation even while the applied force during testing was still in the elastic range. Considering this, it is reasonable to assume that the significant increase in strain toughness, as observed for HA_6 sample, can be ascribable to the intergranular micro-porosity; this may cause more micro-fractures under the same strain and finally an increased strain energy consumption. Moreover, the friction between broken fragments of cell walls dissipated extra energy [41]. Likewise, the higher maximum compressive strength value exhibited by the control sample can be ascribable to the stronger necking areas between the grains, possibly requiring more energy to break the struts than for the HA_6 sample.

## 3. Materials and Methods

### 3.1. Chemicals

Mussel shells (from Mytilus edulis) were kindly furnished by a local fishing company (Lepore Mare SpA, Fasano (BR), Italy). Ammonium phosphate dibasic ((NH_4_)_2_HPO_4_, ≥98%), phosphoric acid (H_3_PO_4_ 85% *w*/*w*), and ammonium hydroxide (NH_4_OH 30% *v*/*v*) used for the synthesis of HA were purchased from Sigma–Aldrich (Milan, Italy). Dolapix CA (Zschimmer and Schwartz, Lahnstein, Germany), Olimpicon A (Olimpia Tensioattivi, Cavenago Brianza (MB), Italy), W53 (Zschimmer and Schwartz, Lahnstein, Germany) were used for the scaffold preparation. Ultrapure water (18.2 MΩ/cm) was used.

### 3.2. Synthesis of CaO Samples 

Waste mussel shells were manually cleaned with water, dried overnight at 80 °C, and mechanically crushed. After this, they were heated in a muffle furnace mod. ZA (Prederi Vittorio & Figli, Milan, Italy) to obtain CaO according to the reaction shown in Equation (1).
(1)(Mussel ShellsCaCO3−sources)→T=700−1000 °CHeating rate=5 °C/minAir/N2 CaO+CO2

Different treatments were performed, changing some parameters, such as temperature (700 or 1000 °C) and atmosphere (air or in N_2_), as reported in Table 1. All the thermal treatments were performed by fixing a heating rate of 5 °C/min and a swell time of 1 h. Powders were cooled naturally, manually milled, and mechanically sieved (Retsch GmbH, Haan, Germany) under 125 μm mesh size.

### 3.3. Synthesis of Hydroxyapatite Samples

The synthesis of HA was performed using two different CaO based powders derived from mussel shells. Two different phosphorus sources, (NH_4_)_2_HPO_4_ [42] or H_3_PO_4_ [43], were employed to synthesize HA according to the reactions showed in Equations (2) and (3), respectively.
(2)$$10Ca{\left(OH\right)}_{2}+6{\left(NH_{4}\right)}_{2}HPO_{4}\to Ca_{10}{\left(PO_{4}\right)}_{6}{\left(OH\right)}_{2}+6H_{2}O+12NH_{4}OH$$
(3)$$10Ca{\left(OH\right)}_{2}+6H_{3}PO_{4}\to Ca_{10}{\left(PO_{4}\right)}_{6}{\left(OH\right)}_{2}+18H_{2}O$$

In detail, 2 g of S_A_700 or S_A_1000 were dispersed in 35 mL of ultrapure water, causing the complete transformation of CaO into Ca(OH)_2_; this was followed by a slow dropping of 0.6 M aqueous solution of (NH_4_)_2_HPO_4_ or H_3_PO_4_. When H_3_PO_4_ was used as the phosphorus source, the pH of the reaction was adjusted to 9.0 by adding NH_4_OH, while, in both cases, the temperature and the incubation time were set at 80 °C and 16 h, respectively. Then, the powders were filtered, washed several times with ultrapure water by centrifugation at 4000 rpm for 15 min and freeze-dried overnight. The methods of preparation of the HA powders are reported in Table 2.

#### Scale-Up Synthesis of Hydroxyapatite Sample

Based on the results of the previous section, the sample HA_1 was chosen for the scaffold preparation. Because of this, the volume of the reaction mixture for the preparation of HA_1 was scaled up by 100 times to a pilot scale. The reaction was carried out in a mechanically stirred reactor (capacity 30 L).

The reaction was carried out at 80 °C by suspending 420 g of S_A_700 powder in 7.5 L of ultrapure water, to prepare a 1.0 M aqueous suspension of Ca(OH)_2_; then 7.5 L of 0.6 M (NH_4_)_2_HPO_4_ was added drop-wise at 1 mL/min under continuous stirring. The mixture was left stirring at 80 °C for 16 h; after this, the powder suspension was cooled at room temperature, recovered and washed by filtration with ultrapure water and lyophilized overnight. The obtained powder (labeled as HA_5) was crushed and sieved under 125 μm mesh size.

### 3.4. Preparation of the 3D Scaffolds

The HA_5 sample was calcined at 1000 °C for 1 h and sieved under 150 μm, to obtain the sample (labeled as HA_6) used to produce macroporous scaffolds according to an already reported direct foaming process [25]. As a reference sample, scaffolds based on commercial HA powders (Sigma–Aldrich, Milan, Italy) were also prepared. Briefly, the powders (HA_6 and commercial HA calcined at 1000 °C) were dispersed in water with Dolapix CA, according to the weight ratio HA:H_2_O:Dolapix = 73:23:4. Then, 2 wt% of Olimpicon A and 0.7 wt% of W53, with respect to the powder, were added as foaming agents. Such foamed suspensions were finally poured into laboratory filter paper (weight 60 g/m^2^, thickness 0,13 mm) molds, dried at 25 °C for 24 h, and sintered at 1250 °C for 1 h. The schematic representation of the process is reported in Figure 2. 

### 3.5. Characterizations of the Materials

X-Ray Diffraction (XRD) analysis was performed using a DS Advance Diffractometer (Bruker, Karlsruhe, Germany), equipped with a Lynx-eye position-sensitive detector, with CuKα radiation (λ = 1.54178 Å), at 40 kV and 40 mA. XRD patterns were acquired in the 20–60° (2θ) at a step size of 0.02 °/step, and a scanning speed of 0.5 s/step. Phase identification was performed through Rietveld refinement with the software TOPAS5 [44]. The weight composition of the phases was refined considering a multiphase system, using tabulated atomic coordinates of HA (ASTM Card file No. 09-0432), CaO (ASTM Card file No. 37-1497), Ca(OH)_2_ (ASTM Card file No. 04-0733), and calcite (ASTM Card file No. 05-0586). Symmetrized spherical harmonics were used to consider the anisotropic peak broadening effects due to anisotropic crystal shape. The average size of crystal domains along two directions (D_(002)_) and (D_(310)_) were calculated applying Scherrer’s equation (Equation (4)) [45]:(4)$$D_{\left[hkl\right]}=\frac{0.9\lambda }{cos\theta \sqrt{\left(\Delta_{r}^{2}\right)-\left(\Delta_{0}^{2}\right)}}$$
where θ is the diffraction angle for the plane (hkl), Δ_r_ and Δ_0_ are the widths in radians of the reflection hkl at half height for the synthesized and pure inorganic hydroxyapatite (standard reference material, calcium hydroxyapatite, National Institute of Standards and Technology), respectively, and λ = 1.5405 Å. 

Scanning Electron Microscopy (SEM) was performed with a Sigma NTS equipment (Zeiss, Jena, Germany). Samples were sputtered with gold with a Sputter Coater E5100 (Quorum Technologies, former Polaron Unlimited, Lewes, UK) before the analysis. Transmittance Electron Microscope (TEM) was done with low-resolution JEM-1011 microscopy (Jeol, Tokyo, Japan) operating at 100 kV equipped with a CCD camera ORIUS 831. The samples were prepared by dropping a dilute solution of each powder onto carbon-coated copper grids, then allowing the water to evaporate. Finally, the grids were rapidly transferred to the microscope.

The elemental composition was carried out by inductively coupled plasma atomic emission spectrometer (ICP-OES) (Agilent Technologies, Santa Clara, CA, USA). Samples were prepared by dissolving 5 mg of powder in 50 mL of 1 wt% HNO_3_ solution. The most abundant elements among those detected were quantified using the following wavelength: Ca 422.673 nm, Mg 279.553 nm, Na 588.995 nm, P 213.618 nm, Se 207.479 nm, and Sr 421.552.

Specific surface area (SSA) of the powdered samples was measured through N_2_ gas adsorption modeled by the Brunauer–Emmett–Teller (BET) method. The BET N_2_ gas adsorption method was employed using a Surfer instrument (Thermo Fisher Scientific, Waltham, MA, USA). The measurement error is related to the accuracy of N_2_ adsorption/desorption techniques (<1%). 

Thermogravimetry analyses (TGA) were performed using an STA 449F3 Jupiter (Netzsch GmbH, Selb, Germany) apparatus. About 10 mg of sample was weighed in an alumina crucible and heated from room temperature to 1100 °C under air flow with a heating rate of 10 °C/min.

The particles surface charge of the powders, expressed in terms of ζ-potential, were analyzed using the electrophoretic mobility measurement on a Zetasizer Nano analyzer (Malvern, Worcestershire, UK), by suspending the samples (0.5 mg/mL) in water with 4 wt%. of Dolapix CA. The particle size measurements were performed on the same samples and with the same equipment used for the evaluation of particles surface charge, by dynamic light scattering (DLS) with backscatter detection (λ = 630 nm; θ = 173°); the results are reported as Z-average of hydrodynamic diameters and relative polydispersity indexes of four measurements of at least 10 runs for 10 s at 25 °C.

The relative density of the scaffolds was determined by the ratio ρ_c_/ρ_s_; ρ_c_ is the density of the material calculated as weight-on-volume ratio, while ρ_s_ is the theoretical density of the phase, determined considering the composition calculated by XRD data. Then, the porosity of the structure was evaluated as ϕ = 1 − ρ_c_/ρ_s_. The compressive strength was determined by testing 10 parallelepiped specimens (base = 9 × 9 mm, height = 18 mm). The tests were performed in displacement control at 0.5 mm/min with a universal testing machine (Zwick/Roell Z050, Ulm, Germany). Young’s modulus was calculated as the slope of the stress–strain curves in the elastic region. The strain energy or toughness, defined as the energy required to fracture the specimen, was also calculated by measuring the area underneath the compressive stress–strain curve. 

## 4. Conclusions

HA nanoparticles were successfully prepared from mussel shells (*from Mytilus edulis*), which are one of the most abundant by-products produced by fish industries. The results show that the physicochemical properties of HA powders can be tailored by selecting the appropriate conditions of the process. The final use of the as-obtained HA powders could be different according to their properties/features, including drug delivery, tissue engineering, and environmental remediation. Herein, selected HA powder was effectively used to prepare a macroporous 3D scaffold with open interconnected porosity (87–89%, pore size between 50 and 300 μm) and good mechanical properties (compressive strength of 0.51 MPa) by direct foaming technique. The obtained scaffold has all the physical and mechanical requisites for being suitable to be used as bone substitute. Additional investigations will be carried out in the future to study the in vitro and in vivo behavior of the scaffold as well as the preparation of similar devices with other additive manufacturing technologies, such as 3D printing. Our results demonstrate that mussel shell by-products constitute a sustainable and cost-efficient source usable in a circular economy approach for the development of compounds of high added value in the biomedical field. Indeed, the conversion of seashells into a porous 3D scaffold for biomedical applications can absorb a very small portion of the materials coming from the fishing industry side streams, offering only limited environmental and economic benefits. However, the transformation of these by-products into high added value materials by a green and easily scalable processes, such as that reported in this study, is an example of the impact that the application of the circular economy principles to the fishing industry can have on this and other industrial sectors. This work could pave the way towards the development of new valorization ideas, utilizing mussel shells for diverse applications. 

## Figures and Tables

**Figure 1 marinedrugs-18-00309-f001:**
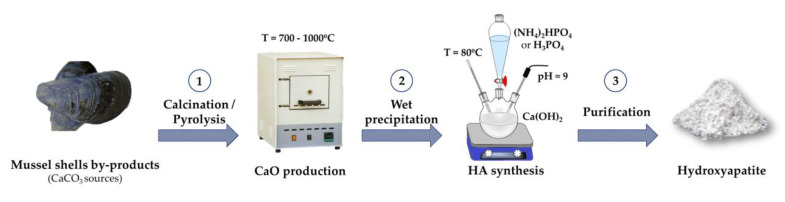
Schematic representation of the process employed in this work to generate hydroxyapatite (HA) powders from mussel shells.

**Figure 2 marinedrugs-18-00309-f002:**
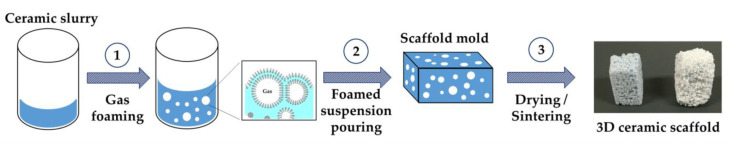
Schematic representation of the direct foaming method to prepare a macroporous 3D HA scaffold.

**Figure 3 marinedrugs-18-00309-f003:**
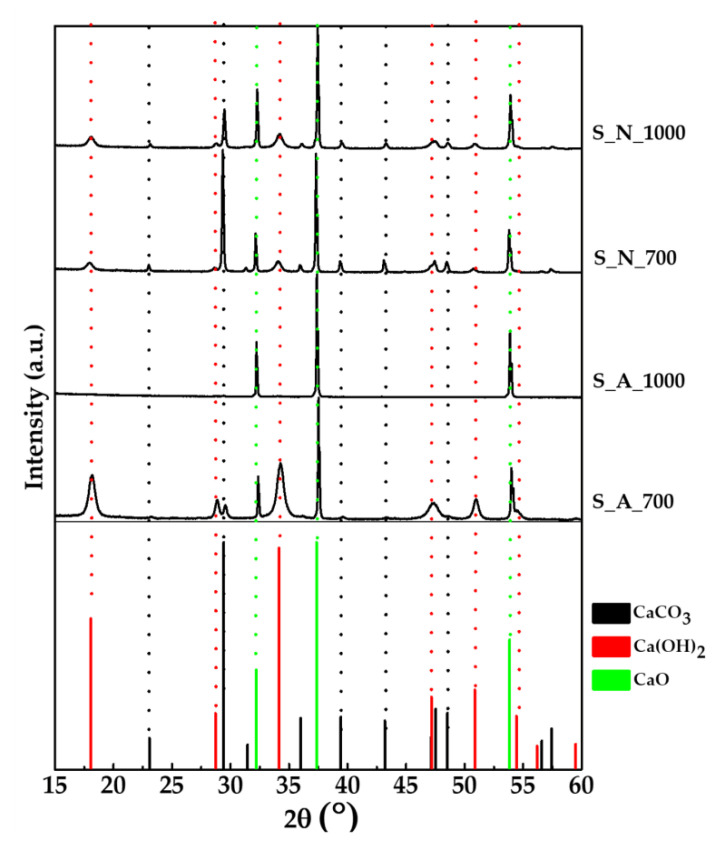
XRD patterns of the CaO samples; reference patterns of CaCO_3_ (ASTM Card file No. 05-0586), CaO (ASTM Card file No. 37-1497), and Ca(OH)_2_ (ASTM Card file No. 04-0733) are also shown.

**Figure 4 marinedrugs-18-00309-f004:**
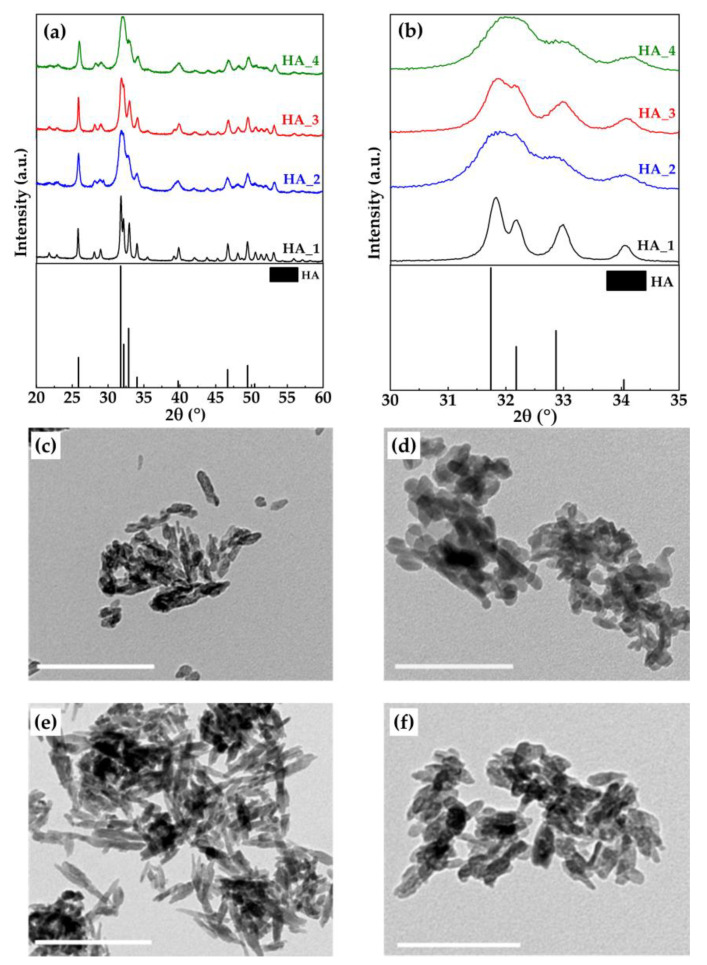
(**a**) XRD pattern of HA samples; XRD pattern of the standard of HA (ASTM Card file No. 09-0432) is also shown; (**b**) enlargement of the 30–35° (2θ) range; (**c**–**f**) TEM micrographs of HA_1, HA_2, HA_3, and HA_4, respectively. Scale bar is 200 nm.

**Figure 5 marinedrugs-18-00309-f005:**
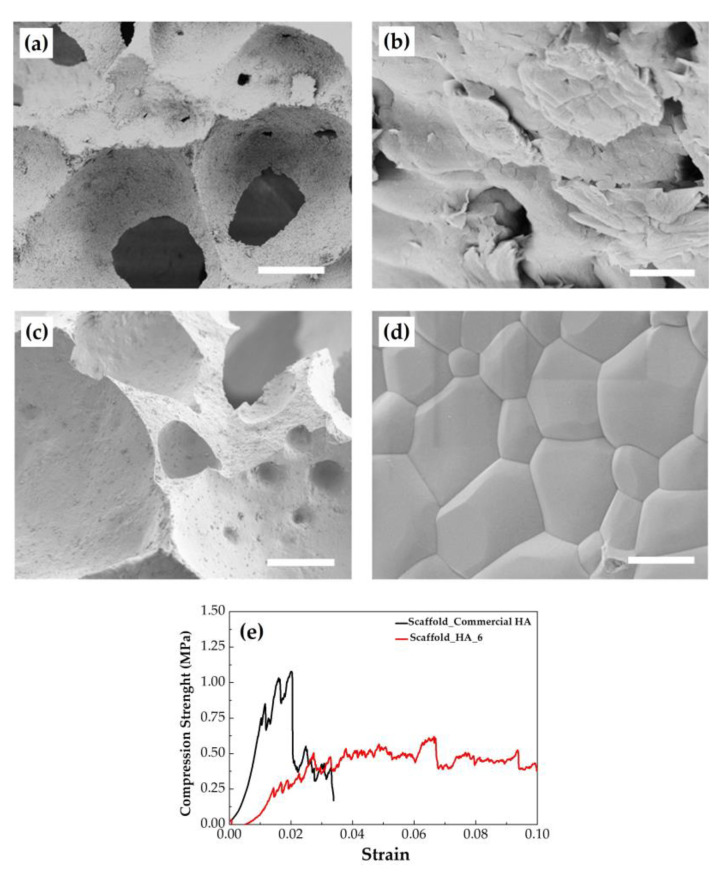
SEM micrographs of scaffolds prepared with (**a**,**b**) HA_6 and (**c**,**d**) commercial HA. Scale bar: (**a**,**c**) 100 μm; (**b**,**d**) 1 µm; (**e**) curve showing the compressive strength vs. strain for both scaffolds.

**Table 1 marinedrugs-18-00309-t001:** CaO samples derived from the mussel shells prepared with different treatments. Corresponding phase composition (wt%) and surface area (m^2^/g) are reported.

Sample Name	Treatment	Temperature (°C)	Phase Composition (wt%)	Specific Surface Area (SSA) (m^2^/g)
S_A_700	Air	700	CaO (19.0 ± 0.3); Ca(OH)_2_ (70.1 ± 0.4); Calcite (10.9 ± 0.4)	5.6
S_A_1000	Air	1000	CaO (100%)	3.1
S_N_700	N_2_	700	CaO (32.0 ± 0.3); Ca(OH)_2_ (17.8 ± 0.4); Calcite (50.2 ± 0.4)	2.5
S_N_1000	N_2_	1000	CaO (43.9 ± 0.3); Ca(OH)_2_ (27.6 ± 0.3); Calcite (28.6 ± 0.4)	6.6

**Table 2 marinedrugs-18-00309-t002:** Hydroxyapatite (HA) samples prepared from calcined mussel shells with different protocols. Corresponding phase composition (wt%), Ca/P molar ratio, and surface area (m^2^/g) are reported.

Sample Name	Ca Containing Reagent	P Containing Reagent	Phase Composition (wt%)	Ca/P Ratio (mol/mol)	Specific Surface Area (m^2^/g)
HA_1	S_A_700	(NH_4_)_2_HPO_4_	HA (100%)	1.88 ± 0.01	49.6
HA_2	S_A_700	H_3_PO_4_	HA (99.0 ± 0.1); Calcite (1.0 ± 0.1)	1.73 ± 0.01	100.9
HA_3	S_A_1000	(NH_4_)_2_HPO_4_	HA (100%)	1.84 ± 0.01	83.5
HA_4	S_A_1000	H_3_PO_4_	HA (100%)	1.73 ± 0.01	93.1

**Table 3 marinedrugs-18-00309-t003:** Crystal domains’ size of HA samples.

Sample Name	D_(002)_ (nm)	D_(310)_ (nm)	D_(002)_/D_(310)_
HA_1	108.7 ± 1.5	55.2 ± 0.6	2.0
HA_2	33.8 ± 0.5	11.9 ± 0.1	2.8
HA_3	70.3 ± 0.6	21.0 ± 0.2	3.4
HA_4	30.2 ± 0.5	12.0 ± 0.2	2.5
HA_5	80.7 ± 0.7	30.6 ± 0.1	2.6

**Table 4 marinedrugs-18-00309-t004:** Phase composition and surface area of HA_5 (labeled as HA_6) and commercial HA powders after calcination at 1000 °C for 1 h.

Sample Name	Phase Composition (wt%)	Specific Surface Area (m^2^/g)
HA_6	HA (96.0 ± 0.1); CaO (4.0 ± 0.2)	4.5
Commercial HA	HA (100%)	5.1

**Table 5 marinedrugs-18-00309-t005:** Mechanical properties of the scaffolds.

Sample Name	Porosity (%)	Compressive Strength (MPa)	Young’s Modulus (MPa)	Work of Fracture (mJ/m^3^)
Scaffold HA_6	87.8 ± 0.1	0.51 ± 0.14	36 ± 12	34 ± 7
Scaffold Commercial HA	85.4 ± 0.7	1.06 ± 0.26	68 ± 26	16 ± 6

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
