# Peer review of "Mussel Shell-Derived Macroporous 3D Scaffold: Characterization and Optimization Study of a Bioceramic from the Circular Economy"

_marinedrugs, 2020, doi:10.3390/md18060309_

Round 1
Reviewer 1 Report
This paper describes the use of a bio-sourced calcium oxide from waste mussel shells to prepare porous hydroxyapatite bodies.
The paper is very well written, constructed and presented. However, some issues exist.
The discussion opens with statements about the importance of creating circular economies and finding ways to utilise what might otherwise be waste materials, that retain intrinsic value, such as mussel shells, as used here.
This is generally true and to be encouraged and an area in which I work. A problem with that argument, in this particular context, is that the proposed technology would only ever utilise a miniscule proportion of the mussel shell produced. However, that only goes to illustrate the magnitude of the problem of turning waste from one industry sector into a useful product in another industry sector, an issue my work also faces. I am of the belief that it behoves the authors to identify such issues to readers, because no work of science exists in isolation.
The authors place significant faith in the notion that crystalline hydroxyapatite is the ideal material for the fabrication of bio-scaffold systems. This has been the subject of significant debate for 30 or more years and continues to be so and my own opinion is that that faith is flawed. But that in no way takes away from the work that has been done here. The work has been well done and of course, the material described may prove to be the right material.
However, in my opinion, an issue exists with the paper in that the only innovation that I can detect, is the use of the mussel shell as the source of calcium oxide. I am not sufficiently conversant with the literature of the use of this particular waste product to know if that is actually innovative.
All of the chemistry and processes used are well established, from calcination of the shell to the conversion of the shell into hydroxyapatite and then the use of typical and also patented ceramic slip techniques to prepare the porous scaffolds.
The calcination and subsequent precipitation chemistry techniques are found in textbooks. Of course that is no problem, if it is stated or discussed that the techniques are textbook techniques.
The authors describe that inert N2 atmosphere is not optimal. The authors do not describe what happens to the organic component of the shell during heating in that inert atmosphere. In air, the organic component will oxidise, but in N2?
The slip techniques used are typical for the preparation of ceramic green bodies prior to sintering.
It seems to me that a potential problem lies in that the substantive technique used for preparing the porous body was patented in 2009 in EP 1411035B1 by a company, with which, I suggest the authors will (should) be familiar. The technique used herein is not substantively different to that work and is, in my opinion, therefore covered. This would be fine, if it were discussed in the paper and said patent were referenced. This is not discussed in the paper, nor is said patent referenced. All of that my be referenced in Reference 25, but I unfortunately do not have access to this work, so cannot know if it is or not.
Furthermore, one of the more important steps in the preparation of the dried green-body has been completely ignored in the paper. That is the process of drying the slip. If this is not performed correctly, the 3D body will not form as it will crack excessively. Much of the preparation method has been described well, this part has not. It should be as it is key.
I remain frustrated constantly by authors who omit important steps of chemistry or process from their work. In this case, to achieve the goal of the intact 3D ceramic body, this is an important step.
In my opinion the work had been well done and described, but it is far from innovative or novel. I am unfamiliar with the standards required by this journal, so cannot comment on its applicability for this journal. But it strikes me that this paper does not describe "the research, development, and production of biologically and therapeutically active compounds from the sea" and might be best suited to a bioceramics, waste materials conversion, or general materials journal.
Author Response
This paper describes the use of a bio-sourced calcium oxide from waste mussel shells to prepare porous hydroxyapatite bodies.
The paper is very well written, constructed and presented. However, some issues exist.
The discussion opens with statements about the importance of creating circular economies and finding ways to utilise what might otherwise be waste materials, that retain intrinsic value, such as mussel shells, as used here.
This is generally true and to be encouraged and an area in which I work. A problem with that argument, in this particular context, is that the proposed technology would only ever utilise a miniscule proportion of the mussel shell produced. However, that only goes to illustrate the magnitude of the problem of turning waste from one industry sector into a useful product in another industry sector, an issue my work also faces. I am of the belief that it behoves the authors to identify such issues to readers, because no work of science exists in isolation.
We thank the reviewer for appreciating our work and, overall, the issue of waste valorisation. We agree with the comments. Sentences to clarify the concepts that this application can absorb a very small portion of the materials coming from the fishing industry side streams as well as that this work is an example of the impact that the application of the circular economy principles to the fishing industry can have on this and on other industrial sectors were added in the conclusions (page 12, lines 445-454).
The authors place significant faith in the notion that crystalline hydroxyapatite is the ideal material for the fabrication of bio-scaffold systems. This has been the subject of significant debate for 30 or more years and continues to be so and my own opinion is that that faith is flawed. But that in no way takes away from the work that has been done here. The work has been well done and of course, the material described may prove to be the right material.
However, in my opinion, an issue exists with the paper in that the only innovation that I can detect, is the use of the mussel shell as the source of calcium oxide. I am not sufficiently conversant with the literature of the use of this particular waste product to know if that is actually innovative.
All of the chemistry and processes used are well established, from calcination of the shell to the conversion of the shell into hydroxyapatite and then the use of typical and also patented ceramic slip techniques to prepare the porous scaffolds.
The calcination and subsequent precipitation chemistry techniques are found in textbooks. Of course that is no problem, if it is stated or discussed that the techniques are textbook techniques.
The main aim of this work was to demonstrate that hydroxyapatite prepared from mussel shells can be used for the generation of 3D macroporous scaffolds. The technique used for the preparation of the scaffolds are not novel and are based on already reported methodologies. This was clearly stated now at page 3 lines 105-106. The direct foaming method has been also selected for the preparation of the scaffolds in a perspective of low economic and environmental impact since it does not use sacrificial templates. Also this part was now clarified with a sentence at page 3 lines 105-107.
The authors describe that inert N2 atmosphere is not optimal. The authors do not describe what happens to the organic component of the shell during heating in that inert atmosphere. In air, the organic component will oxidise, but in N2?
A sentence was added specifying that in inert atmosphere the organic carbon of the shell converts into inorganic graphitic-like carbon (page 5, lines 158-160).
The slip techniques used are typical for the preparation of ceramic green bodies prior to sintering.
It seems to me that a potential problem lies in that the substantive technique used for preparing the porous body was patented in 2009 in EP 1411035B1 by a company, with which, I suggest the authors will (should) be familiar. The technique used herein is not substantively different to that work and is, in my opinion, therefore covered. This would be fine, if it were discussed in the paper and said patent were referenced. This is not discussed in the paper, nor is said patent referenced. All of that my be referenced in Reference 25, but I unfortunately do not have access to this work, so cannot know if it is or not.
A sentence addressing this point and how our work is different from what reported in the patent was added (i.e. the use of ball milling), page 7, lines 253-255. More details of this process are reported in Reference 25.
Furthermore, one of the more important steps in the preparation of the dried green-body has been completely ignored in the paper. That is the process of drying the slip. If this is not performed correctly, the 3D body will not form as it will crack excessively. Much of the preparation method has been described well, this part has not. It should be as it is key.
I remain frustrated constantly by authors who omit important steps of chemistry or process from their work. In this case, to achieve the goal of the intact 3D ceramic body, this is an important step.
The authors agree with the reviewer that the drying of the foamed suspension is surely a critical step if considering the risk of crack generation. In this respect, we prevented flaws generation by pouring the foamed suspension in laboratory filter paper (weight 60 g/m2, thickness 0,13 mm) mold, kept motionless for 24h at 25°C, followed by paper removal and thermal sintering. These details were added in the manuscript (page 8, lines 282-285 and page 11, lines 380-381).
In my opinion the work had been well done and described, but it is far from innovative or novel. I am unfamiliar with the standards required by this journal, so cannot comment on its applicability for this journal. But it strikes me that this paper does not describe "the research, development, and production of biologically and therapeutically active compounds from the sea" and might be best suited to a bioceramics, waste materials conversion, or general materials journal.
One of the subject areas of the journal Marine Drugs is “Biomaterials of marine origin”, moreover this manuscript will be published in a special issue of the journal, dedicated to “Bone Regeneration”, therefore we think that this work falls completely within the scope of the journal.
Reviewer 2 Report
The article is very interesting and fits in with the latest research trends. Research methodology and its implementation is very good. I would recommend the manuscript for publication in Marine Drugs.
Author Response
The article is very interesting and fits in with the latest research trends. Research methodology and its implementation is very good. I would recommend the manuscript for publication in Marine Drugs.
We thank the reviewer for his/her very positive assessment of our work.
Reviewer 3 Report
I have only a question and some remarks:
- Do the authors have elementary composition of mussel shells?
- I think, from a formal point of view of the paper, that font in Figures should be the same size as the text in the manuscript. For example, in Figure 3 the font size of description of x and y axes is too big. Figure 5 – the font of description of 5 panels – (a), (b), (c), (d), (e) is too big as well; the same for supplementary panels of Figures S1-S3.
Author Response
I have only a question and some remarks:
- Do the authors have elementary composition of mussel shells?
The XRD, TGA and ICP-OES analysis of the mussel shells were added in the SI and discussed in the text (page 3, lines 115-122).
- I think, from a formal point of view of the paper, that font in Figures should be the same size as the text in the manuscript. For example, in Figure 3 the font size of description of x and y axes is too big. Figure 5 – the font of description of 5 panels – (a), (b), (c), (d), (e) is too big as well; the same for supplementary panels of Figures S1-S3.
The figures were changed according to the referee’s suggestion.